# Incidence and Risk Factors for Wound Revision after Surgical Treatment of Spinal Metastasis: A National Population-Based Study in South Korea

**DOI:** 10.3390/healthcare11222962

**Published:** 2023-11-15

**Authors:** Han-Dong Lee, Hae-Dong Jang, Jin-Sung Park, Nam-Su Chung, Hee-Woong Chung, Jin-Young Jun, Kyungdo Han, Jae-Young Hong

**Affiliations:** 1Department of Orthopaedic Surgery, Ajou University School of Medicine, Suwon 16499, Republic of Korea; handonglee@gmail.com (H.-D.L.); ajouosns@gmail.com (N.-S.C.); 110511@aumc.ac.kr (H.-W.C.); mogulin@aumc.ac.kr (J.-Y.J.); 2Department of Orthopaedic Surgery, Soonchunhyang University Bucheon Hospital, Bucheon 14584, Republic of Korea; khaki00@schmc.ac.kr; 3Department of Orthopedic Surgery, Spine Center, Samsung Medical Center, Sungkyunkwan University School of Medicine, Seoul 06351, Republic of Korea; paridot@daum.net; 4Department of Statistics and Actuarial Science, Soongsil University, Seoul 06978, Republic of Korea; hkd917@naver.com; 5Department of Orthopedics, Korea University Hospital, Ansan 15355, Republic of Korea

**Keywords:** spine neoplasms, surgical wound infection, surgical wound dehiscence, surgical procedures, operative

## Abstract

Wound complications are commonly seen after surgeries for metastatic spine tumors. While numerous studies have pinpointed various risk factors, there is ongoing debate. Therefore, this study aimed to verify various factors that are still under debate utilizing the comprehensive Korean National Health Insurance Service database. We identified and retrospectively reviewed a cohort of 3001 patients who underwent one of five surgical treatments (corpectomy, decompression and instrumentation, instrumentation only, decompression only, and vertebroplasty) for newly diagnosed spinal metastasis between 2009 and 2017. A Cox regression analysis was performed to determine the risk factors. A total of 197 cases (6.6%) of wound revision were found. Only the surgical method and Charlson comorbidity index were significantly different between the group that underwent wound revision and the group that did not. Regarding surgical methods, the adjusted hazard ratios for decompression only, corpectomy, instrumentation and decompression, and instrumentation only were 1.3, 2.2, 2.2, and 2.4, with these ratios being compared to the vertebroplasty group (*p* for trend = 0.02). In this regard, based on a sizable South Korean cohort, both surgical methods and medical comorbidity were found to be associated with the wound revision rate among spinal surgery patients for spinal metastasis.

## 1. Introduction

The spine is a predominant site for solid cancer metastasis, with reports indicating that about one third of patients diagnosed with cancer exhibit spinal metastasis [1]. Such metastases to the spine often result in severe pain, a potential risk of paralysis, and significantly impair the patient’s quality of life [2]. Even with the latest progress in drug and radiation therapy, surgery remains pivotal in enhancing treatment outcomes for patients with metastatic tumors. Wound problems are one of the most common surgical complications [1,3,4]. Wound complications severely reduce patients’ quality of life and are the most common cause of increased hospitalization and reoperation [1,5]. Therefore, in an effort to prevent wound complications, various risk factors have been reported.

It has been reported that various surgical-related factors contribute to an increased risk of wound complications, and particularly, the risk increases as the extent of the surgery becomes more extensive [2,6,7]. While there are cases where more extensive surgery is necessary, accepting the risk of wound complications such as paralysis, there is also deliberation in cases of asymptomatic nerve compression as to whether to make the surgery more extensive by adding decompression. There was a study that reported an increase in wound risk with the addition of decompression, yet the issue remains debated [4]. Vertebroplasty is a very simple procedure used for the purpose of pain reduction in spinal tumors. While past studies that proved the efficacy of vertebroplasty in spinal metastatic tumors have reported a 0% risk of wound revision [8,9], there have been case reports of wound complications even after simple vertebroplasty [10,11].

Preoperative and postoperative radiation therapy is also one of the major risk factors for wound revision [7,12]. Theoretically, it can impede wound healing by causing tissue fibrosis and impairing blood circulation, which may increase the risk of wound complications [13,14]. However, there are studies suggesting that radiation therapy does not increase the risk of wound complications [15].

Among the preoperative factors, the one most frequently reported is the patient’s overall condition. Some studies have demonstrated a correlation between wound complications and assessments such as the Karnofsky Performance Scale (KPS) or the Eastern Cooperative Oncology Group (ECOG) performance status, which evaluate a patient’s ability to care for themselves, their daily activity, and their physical ability [2,6]. However, the patient’s condition can be temporary, and these evaluations can be subjective [16].

Therefore, this study aimed to verify various factors that are still under debate using a large cohort. Additionally, we sought to evaluate the patients’ general condition using the relatively objective Charlson comorbidity index (CCI) [17] and to determine its correlation with wound complications. Our hypothesis was that the addition of decompressive surgery during the operation, post-operative radiotherapy, and an increase in CCI scores could elevate the incidence of wound complications in patients with spinal metastatic tumors.

## 2. Materials and Methods

### 2.1. Design

This was a nationwide retrospective cohort study, with data from 2019. The study protocol was approved by the K-NHIS Institutional Review Board. Informed consent was not required since the K-NHIS data are anonymized. The Institutional Review Board of Ajou University Hospital authorized this study (approval no. AJIRB-MED-EXP-21-686).

### 2.2. Population and Sample

We defined a population-based cohort using the claims databases of the Korean National Health Insurance Service (K-NHIS) managed by the Korean government. This covers approximately 97% of the total population, and the medical claims of the remaining 3% of the population are covered by the Medical Assistance Program. The K-NHIS claims databases include extensive information from all clinics and hospitals in Korea regarding diagnoses and comorbidities (coded using the 10th revision of the International Statistical Classification of Diseases and Related Health Problems (ICD-10() [18], demographic characteristics, prescriptions, medical services (treatments and procedures), and the associated costs for both inpatients and outpatients.

### 2.3. Inclusion Criteria

In this study, we included adult patients aged 20 years and above who underwent surgery for newly diagnosed spinal metastasis between 2009 and 2017. We defined a cohort of patients with newly diagnosed spinal metastasis using ICD-10 codes specific to metastatic spine tumors. These ICD-10 codes were C79.5 (secondary malignant neoplasm of bone and marrow) and M49.50 (metastatic fracture of vertebra, multiple sites on the spine). We excluded patients who had been diagnosed with spinal metastasis using the same codes in the previous year. Ultimately, 30,603 patients with newly onset spinal metastasis were enrolled in the study.

Patients who underwent surgery for spinal metastasis were identified using the Korea Informative Classification of Diseases procedural codes (Appendix A) [19]. Surgical procedures were categorized into four major groups: decompression only, instrumentation only, decompression with instrumentation, and a combination of instrumentation, partial, or total corpectomy and palliative vertebroplasty (which includes both vertebroplasty and kyphoplasty). If multiple surgical codes were applied to a patient, the procedure was classified based on the more invasive surgery. Ultimately, the study cohort comprised 3001 patients who underwent surgical intervention for spinal metastasis.

### 2.4. Group Allocation

Postoperative wound revision was characterized by participants who were assigned both a wound infection code and a suture procedural code within the six months following surgery. Wound infection was identified using ICD-10 codes: M46 (other inflammatory spondylopathies) and T81 (complications of procedures, not classified elsewhere). The procedural codes from the Korea Informative Classification of Diseases for sutures were as follows: M00x (incision), M01x (suture), S00x, SA0x (incision and suture including face and neck), SB0x, and SC0x (incision and suture other than face and neck). Groups that met the criteria for this wound revision definition were designated as the ‘wound revision group’, while those that did not were defined as the ‘control group’.

The precise timeframe post-surgery to classify an infection as a surgical wound infection in patients with spinal metastatic tumors remains undefined. A previous study considered infections occurring up to six months after spinal metastasis tumor surgery as surgical wound infections. This study adopted the same timeframe [3]. Due to the short life expectancy associated with metastatic spine disease, death often limited the scope of follow-up and potentially affected the analysis results. To ensure comprehensive reporting and reduce the risk of underrepresenting wound revision cases, participants who passed away within six months post-surgery were also incorporated into the study. Death was determined by the absence of medical service utilization for over six months [20].

### 2.5. Covariates

The baseline characteristics considered in this study pertained to the surgery and encompassed age (grouped as 20–39, 40–64, and ≥65 years), sex, Charlson comorbidity index (CCI), income, presence of cord compression, and both preoperative and postoperative radiation and chemotherapy. The CCI was derived from age and the historical presence of 17 diseases (Appendix A). These diseases comprised myocardial infarction, congestive heart failure, peripheral vascular disease, cerebrovascular disease, chronic pulmonary disease, rheumatic disease, peptic ulcer disease, mild liver disease, diabetes without chronic complications, diabetes with chronic complications, hemiplegia or paraplegia, renal disease, moderate or severe liver disease, and AIDS/HIV.

Individuals with an income below the 25th percentile were classified as having a low income [21]. Cord compression was identified using ICD-10 codes: M439, M485, M495, G952, G958, G550, G558, and G992. Definitions for radiation and chemotherapy were based on the Korea Informative Classification of Diseases procedural codes, specifically KK151-159 for radiation and HD for chemotherapy. Both radiation and chemotherapy were verified before and after surgery. If administered within three weeks following surgery, they were designated as early postoperative radiation and chemotherapy [4].

### 2.6. Data Analysis

Data were presented as means ± standard deviations for continuous variables and as counts (percentages) for categorical variables. The chi-square test was employed to compare categorical variables, and the student’s *t*-test was used for comparing continuous variables between the wound revision group and the control group. Comparisons among surgical techniques were likewise performed using a consistent approach.

Subsequently, the Kaplan–Meier analysis with a log-rank test and Cox proportional hazard models were utilized for the survival analysis. Hazard ratios (HRs) with 95% confidence intervals (CIs) were computed to determine the risk of wound revision by surgery type. Three models were constructed to investigate the covariates potentially linked with wound revision: Model 1 was unadjusted; Model 2 was adjusted for age and sex; and Model 3 was further adjusted for low income and CCI.

To explore the effects of clinical conditions on the relationship between surgery type and the risk of wound revision, the HRs for wound revision in various subgroups were determined using Cox’s regression analysis, alongside interaction *p*-values. A stratified subgroup analysis by factors including sex, age, cord compression, early postoperative radiation, and chemotherapy was conducted.

All statistical analyses were performed using SAS software (version 9.3; SAS Institute, Cary, NC, USA). A two-sided *p*-value < 0.05 was considered to indicate statistical significance.

## 3. Results

### 3.1. Baseline Characteristics

The mean age of the 3001 patients who underwent surgery for spinal metastasis was 63.3 ± 11.9 years. By age group, 105 people (3.5%) were under 40 years, 1420 people (47.3%) were 40–64 years old, and 1476 people (49.2%) were over 65 years. Men comprised 1948 (64.9%) participants. The mean CCI was 7.7 ± 2.1. The five most common primary cancers were lung (*n* = 800, 26.7%), liver (*n* = 463, 15.4%), colorectal (*n* = 421, 14.0%), prostate (*n* = 413, 13.8%), and breast (*n* = 210, 7%). Approximately 25% of patients (*n* = 757) belonged to the low-income group. Cord compression was confirmed in 25.6% (*n* = 264) of patients.

Surgery was performed in 307 patients (10.2%) with decompression only, 194 patients (6.5%) with instrumentation only, 555 patients (18.5%) with both decompression and instrumentation, 1385 patients (46.2%) with corpectomy, and 560 patients (18.7%) with vertebroplasty. Preoperative radiotherapy and chemotherapy were administered to 1025 patients (34.2%) and 1184 patients (39.5%), respectively. Within three weeks after surgery, 869 patients (29.0%) underwent early radiotherapy and 345 patients (11.5%) underwent early chemotherapy. In total, 1416 patients (47.2%) died within the follow-up period of six months, as shown in Table 1.

### 3.2. Incidence and Risk Factors of Wound Revision

There were 197 cases (6.6%) where wound revision was performed within 6 months after spine metastasis surgery. Among the 3001 patients who underwent surgery, 197 were in the wound revision group, and 2804 were in the control group.

Between these groups, no significant differences in age, sex, income, postoperative radiation and chemotherapy, or spinal cord compression were noted (all *p* > 0.05). However, there were significant differences observed in the CCI and the surgical method. Regarding the surgical method, the frequency of wound revision was lowest in the vertebroplasty group and then in the decompression-only group. The frequency of wound revision was higher in the instrumentation-only, decompression and instrumentation, and corpectomy groups compared to the aforementioned two groups, with similar rates observed among these three groups. The wound revision group had a higher mean CCI (7.9 ± 2.0) compared to the group without wound revision (7.6 ± 2.2, *p* = 0.047), as shown in Table 1.

No significant difference was observed between the wound revision and control groups concerning the proportion of patients who died within the six-month follow-up period.

### 3.3. Wound Revision Risk Factors According to the Surgical Method

There were differences in age and gender among the different surgical method groups. Among individuals aged 65 years or older, corpectomy was less frequently performed, whereas vertebroplasty was more common. Instrumentation-only and vertebroplasty procedures were prevalent in women, whereas decompression-only and decompression combined with instrumentation were more frequent in men.

Postoperative radiotherapy was less common in the vertebroplasty group. There were no statistically significant differences between the groups regarding preoperative radiotherapy, preoperative spinal cord compression, and the CCI, as shown in Table 2.

In a survival analysis adjusted for factors such as age, sex, income, and the CCI, the IRs (incidence rates) of wound revision for the vertebroplasty, decompression-only, instrumentation-only, decompression with instrumentation, and corpectomy groups were 3.2%, 4.6%, 8.2%, 7.4%, and 7.8%, respectively. When considering surgical methods, the adjusted hazard ratios (HRs) with 95% confidence intervals (CIs) for the decompression-only, corpectomy, decompression with instrumentation, and instrumentation-only procedures—when compared to the vertebroplasty group—were 1.3 (0.7, 2.7), 2.2 (1.2, 3.8), 2.2 (1.3, 3.6), and 2.4 (1.2, 4.7), respectively, as shown in Table 3. There was a significant trend observed in these results (*p* for trend = 0.02), as shown in Figure 1.

### 3.4. Subgroup Analysis

Multivariable Cox proportional hazards regression analyses adjusted for confounding variables (age, sex, spinal cord compression, pre- and post-operative radiation therapy) were used to estimate the adjusted HRs for wound revision. The incidence of wound revision did not exhibit any interaction with age, sex, spinal cord compression, and pre- and post-operative radiation therapy (all *p* for interactions > 0.05).

## 4. Discussion

To the best of our knowledge, this is the first study to report on the incidence and risk factors for wound revision following surgery for spinal metastasis, using a large nationwide population-based cohort. In this study, we found that the incidence of postoperative wound revision for spinal metastases was 6.6% within the 6 months following surgery. Interestingly, patients with a higher CCI, indicative of numerous underlying diseases, underwent wound revision significantly more frequently. The wound revision rate was lowest for procedures involving vertebroplasty, followed by those that involved only decompression. Conversely, the highest wound revision rate was observed in patients who had instrumentation in the spine.

While past research has explored the incidence and risk factors of wound complications in spinal metastasis tumor surgery, the majority of these studies were limited by being conducted at single institutions with fewer patients [2,4,6,15,22]. As a result, their findings were often heterogenous. Therefore, a large-scale cohort was used in this study. One of the limitations of these research methods is that errors can occur in the inclusion and exclusion of appropriate patients. In a previous study, using a cohort similar to this study, approximately 30,000 patients with metastatic spine cancer were identified over 10 years. Although the study period was different, in our study, a similar number of spinal metastatic cancer patients were identified [23]. In another study, 1677 patients who underwent surgery for spinal metastasis were recorded in a cohort of about 5 years [24]. Considering that our study was conducted over a period roughly twice as long as that of the previous study, and that the actual number of patients included was also approximately double, it can be inferred that the patients included in our study were appropriately selected.

Previous studies have demonstrated that wound complications in surgeries for spinal metastatic cancer exceed those in other spinal procedures [22]. Wound complications remain the most common postoperative complication in patients who have undergone surgery for spinal metastasis, with incidence rates ranging from 6.5% to 17.2% [1,3,4]. Our study reported a wound revision rate of 6.6%, consistent with previous findings. The relatively low infection rate in comparison to past studies might be attributed to the inclusion of vertebroplasty as a surgical method. Another contributing factor could be our focus on cases where a wound revision was undertaken rather than just instances of wound infection. As such, even when wound infections arose, instances where no revision was conducted were omitted, potentially leading to a reduced incidence rate.

Wound complications in patients with spinal metastatic cancer are not only relatively common but can also lead to severe consequences. As a result, earlier studies have sought to identify various preoperative risk factors for infection to prevent these complications. Many factors related to surgery have been associated with infection in patients who underwent procedures for spinal metastatic cancer. The primary risk factors include a long surgical segment, an extended operation duration, and a posterior approach [2,6,7]. In our study, the likelihood of wound revision varied based on both the surgical method chosen and the duration of the procedure. Specifically, the risk was lowest for vertebroplasty, followed by decompression-only procedures. Interestingly, the wound revision rates were consistent across the three surgical methods that employed implants. In this study, the results appeared consistent with previous research. This is because, generally, procedures like vertebroplasty or decompression have shorter operation times and surgical segments.

Some studies have reported that performing decompression alongside instrumentation can increase the risk of wound complications in spinal metastasis patients [4]. However, in our large cohort from this study, no distinct difference in the wound revision rate was observed between the group that underwent only instrumentation and the group that underwent decompression combined with instrumentation.

Regarding vertebroplasty, while it had the lowest wound revision rate, it was not 0%, as previously reported in the literature [8,9]. In some studies, post-vertebroplasty infection in metastasis is reported [10,11]. In this study, consistent with some previous studies, it was demonstrated that vertebroplasty in metastasis can lead to wound revision. In our large cohort from this study, the incidence appeared to be around 3%.

One of the most commonly reported preoperative risk factors is the patient’s overall health. Several studies highlight the link between a patient’s functional performance, indicating their general health, and postoperative wound complications. Tavares-Júnior et al. found that having an ECOG Performance Status of three or four increased the risk of postoperative infections [6]. Similarly, Carl et al. observed that patients with a KPS score of 70 or higher were 67% less likely to need reoperations due to wound issues than those with a score below 70 [2]. However, it is important to note that these performance scales can sometimes be subjective and yield inconsistent outcomes [16].

In an additional measure of a patient’s overall health, the presence of underlying diseases was evaluated as a potential risk factor for postoperative wound complications [4,22]. In our research, we systematically investigated the association between underlying diseases and wound infections using the CCI. The CCI, a method well-recognized in the medical community for assessing comorbidity, assigns fixed weights, ranging from one to six, to 17 distinct diseases, which are then summed for a maximum possible score of 33 [25]. Lakomkin et al. highlighted the CCI’s potential usefulness in predicting unfavorable outcomes in spine tumor surgeries [26]. The CCI has also been reported as a risk factor for common postoperative infection [27]. In this study, the CCI was significantly higher in the wound revision group. The results of this study provide evidence suggesting that the risk of wound complications may be higher in patients with numerous or serious underlying diseases.

Many studies have cited systemic treatments, such as chemotherapy or radiation therapy for cancer, as significant factors contributing to wound complications [3]. Notably, preoperative radiation therapy [7,12] and early postoperative radiotherapy [4] have been identified as having a close association with wound revision. This connection is believed to arise from the fibrosis caused by radiation in wounds, coupled with its suppression of neovascularization, both of which hinder proper wound healing [13,14]. However, recent research challenges these findings, suggesting that radiation, whether administered pre- or post-surgery, does not have a significant effect on wound complications [15]. In our study, while there was a marginally higher infection rate in the group subjected to preoperative radiation, this difference was not statistically significant. Our analysis also determined that postoperative radiation therapy did not considerably influence the wound revision rate. Based on these findings, the effect of radiation treatment before and after surgery on the wound may be insignificant.

Our study possesses several limitations that warrant mention. First, there is potential for inaccuracies in our research. We determined both the presence of spinal metastases in patients who underwent surgical treatment and the subsequent need for wound revision using only diagnosis and treatment codes. Notwithstanding this limitation, our data on the number of patients with spinal metastases align closely with previous studies, and the incidence of postoperative wound complications we observed is consistent with prior findings. Second, our analysis did not account for certain factors like obesity, steroid use, and smoking habits, each of which might be intricately linked with wound complications. Further research, encompassing these variables, will be necessary for a more detailed assessment of wound complications after surgical treatment. Third, the inclusion of deceased patients in our study might introduce a potential underestimation of the incidence rate. Guided by a previous study [15] that also incorporated deceased patients, given the reduced lifespan of those with spinal metastases, we found it appropriate to include such individuals in our analysis. Notably, despite the inclusion of these deceased patients, we observed no substantial difference in infection rates between them and the surviving participants. This suggests that the risk of underestimating the infection rate is minimal. Fourth, the disparity in sample size between the cases and controls in this study was significant, potentially introducing bias. Therefore, we employed a multivariate regression analysis and a sensitivity analysis to re-evaluate the factors that could contribute to the risk of wound revision. Lastly, our study focused on a South Korean cohort, so care should be taken when applying these results to other populations. Studies with participants from various backgrounds are needed to better understand our findings in different settings.

## 5. Conclusions

Our extensive study, utilizing a large South Korean cohort and coding system analysis, has revealed that both surgical methods and medical comorbidity significantly influence the wound revision rate in patients undergoing spinal surgery. Specifically, our data suggest that adding decompression during instrumentation does not increase this risk, which is an important consideration for surgical planning. Furthermore, even procedures like vertebroplasty can lead to wound complications in patients with spinal metastasis, requiring careful consideration in the management. Notably, we found that patients with spinal metastasis who have various underlying conditions are at an increased risk of developing infections, necessitating extra caution. Also, our findings provide reassurance that systemic chemotherapy and radiation therapy do not adversely affect the wound revision rate, thus supporting their continued use in managing spinal metastasis. Moving forward, these insights can inform clinical decision making, though continued research is recommended to further refine our understanding of these complex interactions.

## Figures and Tables

**Figure 1 healthcare-11-02962-f001:**
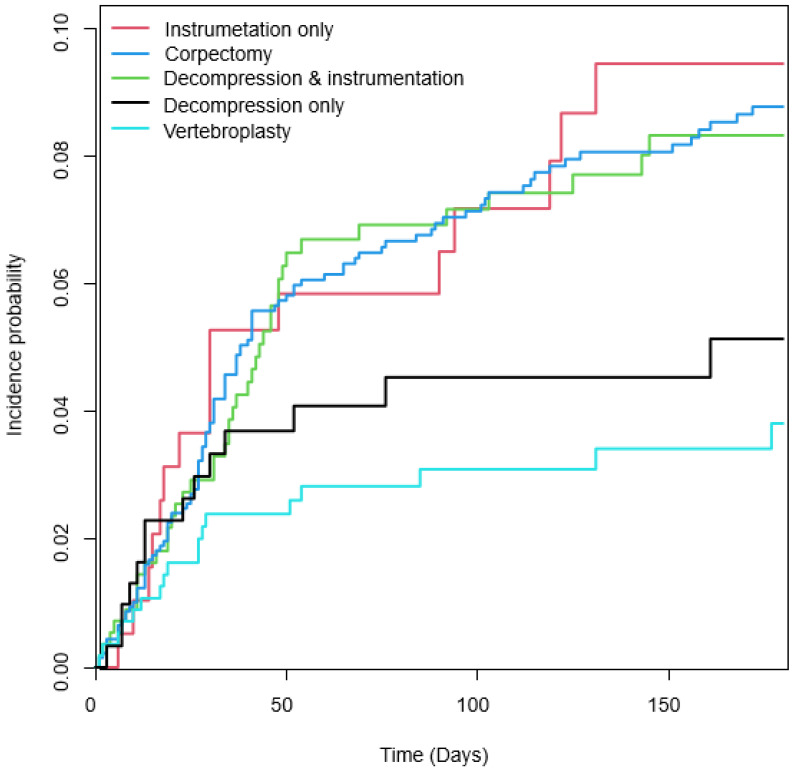
Comparison of the cumulative wound revision according to the surgery for spinal metastasis. The Kaplan–Meier curves with cumulative hazards showed the significantly lower incidence of wound revision in the vertebroplasty and decompression-only groups.

**Table 1 healthcare-11-02962-t001:** Comparison of baseline demographic data between the control group and the wound revision group.

	Total (*n* = 3001)	Wound Revision Group (*n* = 197)	Control (*n* = 2804)	*p*
Age				0.1076
–39	105 (3.5)	11 (5.58)	94 (3.35)	
40–64	1420 (47.32)	100 (50.76)	1320 (47.08)	
65–	1476 (49.18)	86 (43.65)	1390 (49.57)	
Sex, male	1948 (64.91)	126 (63.96)	1822 (64.98)	0.772
Income, low 25%	757 (25.22)	56 (28.43)	701 (25)	0.2845
Surgery				0.0018 *
Decompression only	307 (10.23)	14 (7.11)	293 (10.45)	
Instrumentation only	194 (6.46)	16 (8.12)	178 (6.35)	
Decompression and Instrumentation	555 (18.49)	41 (20.81)	514 (18.33)	
Corpectomy	1385 (46.15)	108 (54.82)	1277 (45.54)	
Vertebroplasty	560 (18.66)	18 (9.14)	542 (19.33)	
Preoperative radiation	1025 (34.16)	74 (37.56)	951 (33.92)	0.2967
Early postoperative radiation	869 (28.96)	52 (26.4)	817 (29.14)	0.4123
Late postoperative radiation	1346 (44.85)	83 (42.13)	1263 (45.04)	0.4272
Preoperative Chemotherapy	1184 (39.45)	87 (44.16)	1097 (39.12)	0.1618
Early postoperative chemotherapy	345 (11.5)	23 (11.68)	322 (11.48)	0.9351
Late postoperative chemotherapy	1034 (34.46)	70 (35.53)	964 (34.38)	0.7419
Cord compression	648 (21.59)	42 (21.32)	606 (21.61)	0.9232
CCI	7.65 ± 2.13	7.94 ± 1.95	7.63 ± 2.15	0.0467 *

Abbreviations: CCI, Charlson comorbidity index. * indicates *p* < 0.05; data in parentheses are percentages.

**Table 2 healthcare-11-02962-t002:** Comparison of baseline demographic data between surgical methods.

	Decompression Only (*n* = 307)	Instrumentation Only (*n* = 194)	Decompression and Instrumentation (*n* = 555)	Corpectomy (*n* = 1385)	Vertebroplasty (*n* = 560)	*p*
Age	11 (3.58)	8 (4.12)	17 (3.06)	61 (4.4)	8 (1.43)	<0.0001 *
–39	143 (46.58)	98 (50.52)	267 (48.11)	741 (53.5)	171 (30.54)	
40–64	153 (49.84)	88 (45.36)	271 (48.83)	583 (42.09)	381 (68.04)	
65–	226 (73.62)	116 (59.79)	373 (67.21)	895 (64.62)	338 (60.36)	
Sex, male	89 (28.99)	53 (27.32)	149 (26.85)	346 (24.98)	120 (21.43)	0.0008 *
Income, low 25%	11 (3.58)	8 (4.12)	17 (3.06)	61 (4.4)	8 (1.43)	0.097
Preoperative radiation	87 (28.34)	66 (34.02)	199 (35.86)	461 (33.29)	212 (37.86)	0.056
Early postoperative radiation	95 (30.94)	55 (28.35)	171 (30.81)	425 (30.69)	123 (21.96)	0.002 *
Late postoperative radiation	122 (39.74)	103 (53.09)	240 (43.24)	738 (53.29)	143 (25.54)	<0.0001
Preoperative	99 (32.25)	69 (35.57)	208 (37.48)	548 (39.57)	260 (46.43)	0.0005 *
Chemotherapy	33 (10.75)	21 (10.82)	65 (11.71)	142 (10.25)	84 (15)	0.0577
Early postoperative chemotherapy	94 (30.62)	80 (41.24)	200 (36.04)	509 (36.75)	151 (26.96)	<0.0001 *
Late postoperative chemotherapy	54 (17.59)	35 (18.04)	131 (23.6)	292 (21.08)	136 (24.29)	0.0816
Cord compression	7.74 ± 2.23	7.43 ± 1.94	7.64 ± 2.06	7.58 ± 2.1	7.85 ± 2.28	0.0616
CCI	87 (28.34)	66 (34.02)	199 (35.86)	461 (33.29)	212 (37.86)	0.056

Abbreviations: CCI, Charlson comorbidity index. * indicates *p* < 0.05; data in parentheses are percentages.

**Table 3 healthcare-11-02962-t003:** The wound revision risk according to the surgical methods.

Surgery	Event (*n*)	HR for Model 1	*p*- Value	HR for Model 2	*p*-Value	HR for Model 3	*p*-Value
Decompression only (*n* = 307)	14	1.421 (0.707, 2.858)	0.0076 *	1.354 (0.67, 2.734)	0.0188 *	1.347 (0.667, 2.721)	0.0171 *
Instrumentation only (*n* = 194)	16	2.481 (1.265, 4.866)		2.332 (1.184, 4.592)		2.379 (1.207, 4.687)	
Decompression and instrumentation (*n* = 555)	41	2.255 (1.296, 3.925)		2.154 (1.233, 3.762)		2.159 (1.235, 3.774)	
Corpectomy (*n* = 1385)	108	2.336 (1.418, 3.847)		2.175 (1.31, 3.609)		2.181 (1.313, 3.621)	
Vertebroplasty (*n* = 560)	18	1 (ref.)		1 (ref.)		1 (ref.)	

Abbreviations: HR, hazard ratio. Model 1 was unadjusted. Model 2 was adjusted for age and sex. Model 3 was additionally adjusted for a low income and CCI. * indicates *p* < 0.05.

## Data Availability

Data available from the authors of this publication.

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
