# Peer review of "Incidence and Risk Factors for Wound Revision after Surgical Treatment of Spinal Metastasis: A National Population-Based Study in South Korea"

_healthcare, 2023, doi:10.3390/healthcare11222962_

Round 1
Reviewer 1 Report
Comments and Suggestions for Authors
Dear Authors,
Tyhank you for this interesting article. In order to improve your paper, the following are my suggestions:
Abstract: The first sentence stating living longer among patients diagnosed with cancer as premise for the rising number of surgeries for spinal metastasis seem not logical, otherwise, you need to be able to support this claim. The three claims in the first sentence don't communicate a clear organization of logic.Kindly rephrase.
Introduction:
The first sentence in the introduction does not serve a good connection with the following sentences thereafter. The connection is weak and not provide connection with the next statements. Wither you delete or you change with a sentence that connects well with the ideas presented in the following statements.
Lines 54-55- WQhat do you mean by 'performance' here?
Line 58 - Cite the "previous studies" you are referring to.
Line 60- What is the gap you are trying to address? Is it the limitation of previous studies in terms of sample size? Discuss clearly the specific gaps you want to address in this study.
Provide a succinct information about the condition in Korea related to your study thesis and the gap you identified. Discuss your thesis statement and your study hypothesis.
Materials and Methods;
I suggest that you categorize your subsections accordingly (Design, Population and Sample, Inclusion criteria, Instruments, Data Gathering Procedure, and Data analysis instead of using 'Surgery for metatstasis, Wound Revision', etc.
Using the standard method subsections will allow you to be able to present the methodology appropriately. The current organization is mixed up and is not helpful to the readers.
What is the percentage of patients diagnosed with spinal metastasis included in the study out of the total patients with cancer?
It is not clear as to how many patients with spine metastasis have undergone surgery and were included in the study and from these number how many underwent wound revision. How many were in the control group?
The variation in the sample size for those with WR and without is wide and can be a source of bias or error. Justify how this was addressed in your study.
Results:
After presenting the participants' profile, I suggest that you organize the presentation of your findings in accordance to your aim:
Incidence of wound revision
Provide a clear picture of the incidence before proceeding to comparisons of characteristrics between those with WR and those without WR.
The word 'required' in line 170 does not give a clear condition whether the patient has undergone or not the wound revision.
Risk Factors- properly label the results that showed the risk factors
Conclusion-What can you conclude from the findings? You merely stated again your findings in your conclusion.
Thank you and good luck.
Comments on the Quality of English Language
Needs moderate editing in terms of idea transition inmany paragraphs.
Author Response
Dear Reviewer,
We are grateful to the reviewers for their meticulous review and valuable feedback, which have enabled us to make significant improvements to the paper. Below are our responses to the points raised in the previous round of reviews. Thank you.
1. Abstract: The first sentence stating living longer among patients diagnosed with cancer as premise for the rising number of surgeries for spinal metastasis seem not logical, otherwise, you need to be able to support this claim. The three claims in the first sentence don't communicate a clear organization of logic.Kindly rephrase.
*Answer: Based on your feedback, we have rephrased the abstract.
2. Introduction:
The first sentence in the introduction does not serve a good connection with the following sentences thereafter. The connection is weak and not provide connection with the next statements. Wither you delete or you change with a sentence that connects well with the ideas presented in the following statements.
Lines 54-55- WQhat do you mean by 'performance' here?
Line 58 - Cite the "previous studies" you are referring to.
Line 60- What is the gap you are trying to address? Is it the limitation of previous studies in terms of sample size? Discuss clearly the specific gaps you want to address in this study.
Provide a succinct information about the condition in Korea related to your study thesis and the gap you identified. Discuss your thesis statement and your study hypothesis.
* Answer: Based on your feedback, we have rephrased the introduction.
3. Materials and Methods;
I suggest that you categorize your subsections accordingly (Design, Population and Sample, Inclusion criteria, Instruments, Data Gathering Procedure, and Data analysis instead of using 'Surgery for metatstasis, Wound Revision', etc.
Using the standard method subsections will allow you to be able to present the methodology appropriately. The current organization is mixed up and is not helpful to the readers.
*Answer: Based on your feedback, we have modified the titles of subsections.
4. What is the percentage of patients diagnosed with spinal metastasis included in the study out of the total patients with cancer?
*Answer: We have not identified patients who underwent surgery for metastatic cancer among all cancer patients; instead, we have only confirmed the codes for patients who underwent surgery for metastatic cancer in the entire population, so we cannot determine the proportion at this time.
5. It is not clear as to how many patients with spine metastasis have undergone surgery and were included in the study and from these number how many underwent wound revision. How many were in the control group?
*Answer: We have added the relevant information to the results section for ease of viewing. Additionally, it can be confirmed in Table 1.
6. The variation in the sample size for those with WR and without is wide and can be a source of bias or error. Justify how this was addressed in your study.
*Answer: We have mentioned this content in the limitations section and have added our response to it.
7. Results:
After presenting the participants' profile, I suggest that you organize the presentation of your findings in accordance to your aim:
Incidence of wound revision
Provide a clear picture of the incidence before proceeding to comparisons of characteristrics between those with WR and those without WR.
The word 'required' in line 170 does not give a clear condition whether the patient has undergone or not the wound revision.
*Answer: We have made revisions to clearly present the incidence of wound revision and the distribution among groups.
8. Risk Factors- properly label the results that showed the risk factors
*Answer: We have made revisions to clearly present the risk factors for wound revision.
9. Conclusion-What can you conclude from the findings? You merely stated again your findings in your conclusion.
*Answer: We have revised the conclusion to more clearly highlight the results and significance of this study.
Thank you and good luck.
Reviewer 2 Report
Comments and Suggestions for Authors
Thank you for allowing me to review this manuscript. Below, I provide a series of recommendations:
- In line 115, include in the age range <40 as 20-40 years.
- In lines 118 – 122, if the medical diagnoses are already indicated in the tables, it is not necessary to mention them in the text.
- In the methods section, include the calculation of effect size when conducting bivariate analyses.
- Consolidate section 3.1. Baseline Characteristics into a table to organize the results. Sections 3.1 and 3.2 could also be combined into a single table, with a column for each group and another for the total sample, along with the p-values.
- Include the effect size of bivariate analyses in the tables as indicated.
- In Table 5, the adjustment indicators for the three calculated models should be specified.
- Consider including in the discussion section the implications for clinical practice.
Author Response
Dear Reviewer,
We are grateful to the reviewers for their meticulous review and valuable feedback, which have enabled us to make significant improvements to the paper. Below are our responses to the points raised in the previous round of reviews. Thank you.
- In line 115, include in the age range <40 as 20-40 years.
* Answer: We have made the modifications you pointed out.
- In lines 118 – 122, if the medical diagnoses are already indicated in the tables, it is not necessary to mention them in the text
* Answer: We have removed the table from the main text and changed it to a supplement.
- in the methods section, include the calculation of effect size when conducting bivariate analyses.
- Include the effect size of bivariate analyses in the tables as indicated.
* Answer: In this study, the effect size of each variable was calculated as the hazard ratio. The section describing the method and statistics was moved because the part about calculating the hazard ratio could be confusing. The table has also been revised to clearly reflect this change.
- Consolidate section 3.1. Baseline Characteristics into a table to organize the results. Sections 3.1 and 3.2 could also be combined into a single table, with a column for each group and another for the total sample, along with the p-values.
* Answer: We have made the modifications you pointed out.
- In Table 5, the adjustment indicators for the three calculated models should be specified.
* Answer: This part has been explained in detail in the statistics section. The original manuscript had a typo that could make it difficult to understand, so it has been corrected.
- Consider including in the discussion section the implications for clinical practice.
* Answer: We have revised the conclusion to more clearly highlight the results and significance of this study.
Reviewer 3 Report
Comments and Suggestions for Authors
Dear authors,
The keywords do not correspond to MeSH terms.
According to STROBE statement, the title or abstract must reflect that it is a cohort study (it is mentioned in the abstract but is not clearly reflected in the methodology of the abstract).
Introduction: correct and appropriate.
Methods: the structure of the method is not adequate in the description of the different sections according to STROBE (study design, setting, participants, variables, data sources/measurement, bias, study size, quatitative variables, statistical methods). For example, the assignment to each of the groups (wound revision group and control group) is not clear.
Tables 1 and 2 can be presented as supplementary materials, while the sociodemographic characteristics would be better described in a table.
The discussion and conclusions are adequate.
Author Response
Dear Reviewer,
We are grateful to the reviewers for their meticulous review and valuable feedback, which have enabled us to make significant improvements to the paper. Below are our responses to the points raised in the previous round of reviews. Thank you.
The keywords do not correspond to MeSH terms.
* Answer: We have modified the keywords to fit the MeSH.
According to STROBE statement, the title or abstract must reflect that it is a cohort study (it is mentioned in the abstract but is not clearly reflected in the methodology of the abstract).
* Answer: We have revised the abstract to more clearly indicate that it is a cohort study.
Introduction: correct and appropriate.
Methods: the structure of the method is not adequate in the description of the different sections according to STROBE (study design, setting, participants, variables, data sources/measurement, bias, study size, quatitative variables, statistical methods). For example, the assignment to each of the groups (wound revision group and control group) is not clear.
* Answer: We have changed the title of the subsection and rearranged the content to better align with the STROBE guidelines.
Tables 1 and 2 can be presented as supplementary materials, while the sociodemographic characteristics would be better described in a table.
* Answer: We have moved Tables 1 and 2 to the supplementary materials.
The discussion and conclusions are adequate.
Round 2
Reviewer 1 Report
Comments and Suggestions for Authors
Dear Authors,
Thank you for the revisions and your efforts.
I only have a few minor suggestions:
Introduction - I suggest to use "patients diagnosed with cancer" more than "cancer patients"
Kindly add the hypothesis of the study at the end of the study aim.
Population and sample- kindly add your reference for the ICD-10
Inclusion Criteria - add reference for the e Korea Informative Classification of Diseases procedural codes
Section 2.4 line 105 - delete the label (definition of wound revision group)
lines 122-123 - support how you determined death
2.5 Covariates - There is an overlap in your age classification (20-40 and 40-64)- is this just in this section or it was extended in the recording of age in your data set?
Add your reference for CCI
Add your reference for income classification
add your basis for designating early post op radiation and chemotherapy
Results - delete table 1 from all the subheadings. Transfer them at the end of each related paragraph. Thesame with the other tables included in all sub headings
Revise the Table titles for Tables 1 and 2. What are you comparing ine ach table? The current titles do not describe what variables are being compared between the two groups.
Thank you and good luck.
Comments on the Quality of English LanguageKindly check for the following:
Subject verb agreement
Sentence construction
Author Response
Thank you once again for your excellent review.
Here are our responses to your comments.
Introduction - I suggest to use "patients diagnosed with cancer" more than "cancer patients"
*Answer: We have made the modification as you suggested.
Kindly add the hypothesis of the study at the end of the study aim.
*Answer: We have included the study's hypothesis at the end of the aim section.
Population and sample- kindly add your reference for the ICD-10
*Answer: We have included the ICD-10 reference.
Inclusion Criteria - add reference for the e Korea Informative Classification of Diseases procedural codes
*Answer: We have included the KICD reference.
Section 2.4 line 105 - delete the label (definition of wound revision group)
*Answer: We have removed the label as you suggested.
lines 122-123 - support how you determined death
*Answer: The study was based on a previous paper that defined death as a lack of use of medical services for a period of six months, and we included the relevant reference to support this finding.
2.5 Covariates - There is an overlap in your age classification (20-40 and 40-64)- is this just in this section or it was extended in the recording of age in your data set?
*Answer: There was indeed an error in the original manuscript, which has now been corrected.
Add your reference for CCI
*Answer: We have included the CCI reference.
Add your reference for income classification
*Answer: We have included the the income classification reference.
add your basis for designating early post op radiation and chemotherapy
*Answer: Based on a prior study reporting that chemotherapy or radiation within three weeks can elevate the risk of wound complications, this definition of 'early' has been established. We included the relevant reference to support this.
Results - delete table 1 from all the subheadings. Transfer them at the end of each related paragraph. Thesame with the other tables included in all sub headings
*Answer: We have made the modification as you suggested.
Revise the Table titles for Tables 1 and 2. What are you comparing ine ach table? The current titles do not describe what variables are being compared between the two groups.
*Answer: We have made the modification.
Reviewer 3 Report
Comments and Suggestions for Authors
Dear authors,
The changes made have improved the quality of the manuscript, although some additional modification is needed.
In the abstract it would be correct to describe the research aims in the same way as in the manuscript text.
Keywords: I reviewed MeSH terms, and for keyword "Spine Neoplasm" the correct MeSH is "Spinal Neoplasms"; for "Surgical wound dehiscence comorbidity" the term MeSH is "Surgical wound dehiscence". For the two keywords "Surgical Procedures" and "Operative" there is a MeSH that includes both concepts in the same descriptor: "Surgical Procedures, Operative"
In the results, do not cite table 1 in the headings of the subheadings. It is more appropriate to describe at the end of each paragraph the following: ...as shown in Table 1. The same for the rest of the tables.
References should include the DOI or link to the primary source.
Author Response
Thank you for your execellent review.
Below are the answers to your comments.
In the abstract it would be correct to describe the research aims in the same way as in the manuscript text.
*Answer: We have changed the abstract as you suggested.
Keywords: I reviewed MeSH terms, and for keyword "Spine Neoplasm" the correct MeSH is "Spinal Neoplasms"; for "Surgical wound dehiscence comorbidity" the term MeSH is "Surgical wound dehiscence". For the two keywords "Surgical Procedures" and "Operative" there is a MeSH that includes both concepts in the same descriptor: "Surgical Procedures, Operative"
*Answer: We have made the modification as you suggested.
In the results, do not cite table 1 in the headings of the subheadings. It is more appropriate to describe at the end of each paragraph the following: ...as shown in Table 1. The same for the rest of the tables.
*Answer: We have made the modification as you suggested.
References should include the DOI or link to the primary source.
* Answer: We have added the dois and links.